# A Narrow Optical Pulse Emitter Based on LED: NOPELED

**DOI:** 10.3390/s22197683

**Published:** 2022-10-10

**Authors:** Diego Real, David Calvo, Antonio Díaz, Francisco Salesa Greus, Agustín Sánchez Losa

**Affiliations:** 1IFIC—Instituto de Física Corpuscular, CSIC—Universitat de València, c/Catedrático José Beltrán, 2, 46980 Paterna, Spain; 2Department of Computer Architecture and Technology/CITIC, University of Granada, 18071 Granada, Spain

**Keywords:** short optical pulse, optical instrumentation

## Abstract

Light sources emitting short pulses are needed in many particle physics experiments using optical sensors as they can replicate the light produced by the particles being detected and are also an important calibration and test element. This work presents NOPELED, a light source based on LEDs emitting short optical pulses with typical rise times of less than 3 ns and Full Width at Half Maximum lower than 7 ns. The emission wavelength depends on the model of LED used. Several LED models have been characterized in the range from 405 to 532 nm, although NOPELED can work with LED emitting wavelengths outside of that region. While the wavelength is fixed for a given LED model, the intensity and the frequency of the optical pulse can be controlled. NOPELED, which also has low cost and simple operation, can be operated remotely, making it appropriate for either different physics experiments needing in-place light sources such as astrophysical neutrino detectors using photo-multipliers or positron emission tomography devices using scintillation counters, or, beyond physics, applications needing short pulses of light such as protein fluorescence or chemodetection of heavy metals.

## 1. Introduction

Many particle physics experiments and applications include optical sensors as detector elements [1,2,3,4,5,6]. The use of devices producing short pulses of light (rise-time < 5 ns) is of great help in these applications, as they could help with the qualification of the light sensors, the calibration of the detectors once in operation, trying to replicate or mimic the light produced by the particles to be detected, or, directly, providing the pulses of light needed to operate the system [7,8,9,10,11]. It is not surprising, as many physics phenomena occur at the nanosecond time scale, and some of them at the picosecond time scale, which requires pulses of the same scale, as well as methods and instruments which can generate the pulses required [12]. State-of-the-art optical pulse generation can achieve pulses with a width of picoseconds, and recently, some pulser based on 2D material saturable absorbers can achieve pulses even below the picosecond [13,14], or pulses with a few picoseconds duration with considerable energy (tenths of nanojoules) [15]. Nevertheless, there are some drawbacks that prevent the use of this kind of optical pulse generator in many applications, such as the wavelength of the pulse generated, usually over 1000 nm, and especially, the complexity of operation and production, with the latest affected by the lack of fine-controlled material fabrication [16]. Many applications do not need such an ultra-fast optical pulse, but instead require lower wavelengths, higher simplicity, and lower cost. LEDs, low-cost silicon devices that produce light by the recombination of the electron-hole [17], can be used for the generation of optical pulses. The disadvantages with respect to laser sources, such as the emission of non-polarized light or the non-uniform intensity distribution, are compensated by their low cost, simple operation, and the possibility of modulating the emission intensity and frequency.

In this work, NOPELED is presented. NOPELED is an instrument that, based on LED technology, can effectively produce short pulses of light, with typical rise times lower than 3 ns and FWHM of about 4–7 ns. The device is inexpensive and has the advantage of modifying the pulse frequency and intensity. Its use can range from photo-multiplier to scintillator counter calibration [18,19,20,21,22,23,24], although not only these as it may help in any application needing an emitter of short pulses of light such as protein fluorescence [25,26], time-resolved optically stimulated luminescence [27], or molecular chemodetection of heavy metals [28,29]. An example of the use of short LED pulses can be found in the time calibration of neutrino telescopes [30], where the photo-multipliers are calibrated using short pulses of light generated by LED instrumentation [31]. An overview of NOPELED is presented in Section 2. In Section 3.1, the pulser circuit is discussed, while the control board is presented in Section 3.2. The layout of both boards is presented in Section 3.3. The embedded software programmed in the microcontroller is described in Section 4, while the results of the qualification of NOPELED are shown in Section 5. Finally, some conclusions are presented in Section 6.

## 2. NOPELED Overview

NOPELED is composed of two electronic boards (see Figure 1), the control board and the pulser board. The control board is responsible for controlling, triggering, and powering up the pulser. The pulser board drives the LED with the voltage and the trigger provided by the control board, generating the short optical pulse. NOPELED is powered at a voltage of 3.3 V with a consumption lower than 50 mA. The operational frequency has been designed to cover the range between 1 and 30 kHz, and the intensity of the emission can be modified by varying the power supply voltage of the pulser board up to 30 V. Either an external trigger or an auto-generated trigger can be used. The architecture of NOPELED is shown in Figure 2. The communication with NOPELED is performed via an Inter-Integrated Circuit (I^2^C) bus connected to the microcontroller. The microcontroller generates the internal trigger signal, the digital signal selects between internal and external triggers, and the Pulse Wave Modulated (PWM) signal controls the voltage to be supplied to the LED.

## 3. Electronics

### 3.1. Pulser Circuit

The pulser is the element of NOPELED where the optical pulse is generated. The NOPELED pulser circuit is based on a design from Kapustinsky [32], which can generate short pulses with basic electronics. The electrical diagram of the pulser is shown in Figure 3. With transistors Q1 (pnp) and Q2 (npn) being in the off state, capacitor C1 loads up to the VLED voltage, the value of which is generated by the control board, and it continues in this value until the transistors are triggered. When the trigger, a 3 V logic signal, arrives, an increase over the base-emitter voltage in transistor Q1 is caused, and the transistor starts to conduct. The conduction of Q1 also makes transistor Q2 conduct, which starts to apply the C1 voltage to the LED, making it flash. As soon as the LED starts to flash, capacitor C1 starts to discharge at a fast rate, which makes transistor Q1 switch off first, and soon after, transistor Q2, bringing the system again to the initial step. In order to sharpen the electrical pulse created, a coil is added in parallel with the LED [33]. It forms a resonant circuit allowing the voltage across the LED to swing negative, and thus the electrical pulse is reduced. The trigger signal is generated and adapted by the control board, which also provides the power supply. The intensity of emission is controlled by the level of the DC voltage. The connection between the controller and the pulser is made by means of three cables, one for the ground, another for the power supply, and the third for the trigger signal.

### 3.2. Control Board

The control board controls the power supply and the trigger to the pulser board. It contains three functional blocks; the control block, the booster, and the trigger block. The control block receives I^2^C communication frames and configures NOPELED. The booster provides the power supply to the LED diode pulser circuit. Moreover, the trigger block generates and supplies the trigger signal to the LED. A 5-pin Molex connector is used for I^2^C communication, the external trigger, and the power signal. The five inputs are: 3.3 V power supply; ground; the I^2^C Communication *SCL* Clock Signal; the I^2^C Communication *SDA* Data Signal; and the external trigger signal.

#### 3.2.1. Control Block

The control subsystem receives the I^2^C commands and configures NOPELED according to the required working mode. This circuit is formed by a Peripheral Interface Controller (PIC) microcontroller. The microcontroller has been configured so that it can work at 3.3 V, which is NOPELED’s power supply voltage and sets the maximum voltage that can be obtained from any output. This type of microcontroller is of the RISC type [34], and the main characteristics are a flash memory with digital communication peripherals, including a Universal Synchronous/Asynchronous Receiver/Transmitter (USART) and a Master Synchronous Serial Port (MSSP) to be used as SPI (Serial Port Interface) or I^2^C. It also has two timers and an 11-channel 10-bit ADC. Through the commands received remotely via I^2^C, the microcontroller generates the control signals both to configure the working modes of the power system and to configure NOPELED. It also reads, via one of the ADC channels, a temperature sensor.

#### 3.2.2. Trigger Block

The internal trigger signal is generated by the microcontroller using one of the PWM outputs. The output provided by the trigger is a squared signal changing from 0 to 3 V, whose frequency can vary from 1 to 30 kHz. The trigger subsystem is able to select between the external trigger signal and the internal one generated by the microcontroller and adapt the received trigger signal. For the trigger selection, five logic gates are used. A microcontroller’s digital output controls the trigger selection. In addition, an ultra-fast comparator has been added to adapt the trigger signal and thus sharpen the rise time of the logic gates. Two main blocks conform to the trigger subsystem:*Selector*: Through five logic gates and one digital selection output of the microcontroller, this block is responsible for selecting between the trigger provided externally and an internal trigger generated by the microcontroller. They are fast response gates; around 4 ns of rise time. The trigger configuration, external or internal, is selected by a microcontroller digital output.*Adapter*: The trigger signal received, whether internal or external, is adapted. A fast comparator improves the response of the output, reducing the rise time of the trigger because the sharper the rise time of the trigger, the shorter the optical pulse produced.

#### 3.2.3. Booster Subsystem

The voltage to set the light intensity of the pulser is provided by the booster. The booster voltage control is generated by the microcontroller. A fixed frequency signal whose duty cycle can be modified after being filtered in a low pass filter controls the output voltage of the micro-converter. The output of the microcontroller can be modified from 0 to 30 V. This circuit also has to supply the necessary power to the control board. The main element is a micro DC/DC converter in a booster configuration. This subsystem is divided into two blocks:*Booster DC/DC:* The electronics necessary for the correct operation of the micro converter is the main component of the booster. The variation in the voltage obtained at the output of the micro converter is obtained by modifying the duty cycle of the signal of the PWM module of the microcontroller. The duty signal varies between 1% and 82% of duty. It is limited by software, being able to modify this value through NOPELED I^2^C communication commands.*Control filter:* The filter adapts the square signal received from the microcontroller module to a signal filtered by a 1-order low-pass filter. This filter will only pass the low frequencies of the signal provided by the PWM module of the microcontroller. In this way, the desired voltage is controlled.

### 3.3. Layout

The dimensions of the NOPELED control board are 48 × 35 mm. The Printed Circuit Boards (PCBs) have been manufactured with a copper thickness of 35 microns and two layers with vias communicating between the TOP and BOTTOM layers. The NOPELED pulser board has a diameter of 20 mm, also with two layers. In both cases, the boards do not present any fabrication difficulty. In Figure 4, the layout (top and bottom) of the control board is shown.

## 4. Embedded Software

The C language has been chosen for programming the microcontroller since, being a high-level language and not associated with any operating system, it is known as the most widely used programming language. The embedded software begins with the configuration of the fuses, ports, and variable initialization. It continues with the loading of the main program where the interruptions are enabled. When the data are received through the I^2^C port, the command is evaluated, and the programmed tasks for each command are performed. The flow diagram of the embedded software is presented in Figure 5. The description of each of the functional blocks follows.

### 4.1. Configuration of Fuses, Serial Ports, and Initialization

In the first step, the configuration bits or “fuses” of the microcontroller are defined, followed by the I^2^C port configuration. I^2^C communication is enabled by hardware. The device is configured as a slave. The address that NOPELED will have is set to “address = 0xa0”. The speed of the I^2^C bus is set to 400 kHz. At a second step, all the variables that are going to be used during the execution of the embedded software are initialized, as well as the subprograms.

### 4.2. I^2^C Interrupt

Every time I^2^C data are received by the microcontroller, the I^2^C interrupt is triggered. At this moment, the data received by the I^2^C port will be stored in the variables created for it, so that once stored, the main program returns at the same point where it was before jumping to the serial interruption.

### 4.3. Main Program

At this stage, the microcontroller is always waiting to receive data through the serial port. Once the I^2^C communication frame has been received, and the data has been saved in the variables created for it, the microcontroller returns to the main program to verify that the received data condition is met, then the “Analyze Frame” function is executed and acts according to the value of the command received. There are four main operation modes defined, and these are:

*Mode 01: “Frequency Mode”.* It generates a pulse train used as an internal trigger. The signal frequency is indicated by the union of the third and fourth received bytes, the third byte being the highest weight and the fourth the lowest weight. The frequency is set between 1 and 30 kHz.

*Mode 02: “Duty Mode”.* It generates a train of pulses with a fixed frequency, but the duty of the signal can be modified as indicated in the third byte received. The values of the duty will vary between 0% and 82%. The LED voltage is controlled by this mode.

*Mode 03: “External Trigger Mode”.* It enables the trigger signal received from outside. This signal is used to trigger the LED pulser.

*Mode Default: “No Broadcast Mode”.* In this mode, the two PWM modules are turned off. In this way, no signal is emitted, and the LED does not flash.

### 4.4. Simulation

The simulation of the I^2^C communication was performed before the production of the first prototypes in order to reduce the development time. The Proteus Virtual Terminal, which is a virtual instrument that tries to emulate the Windows Hyperterminal, has been selected for the simulation. An ISIS software component called “COMPIM” has also been used. It creates a virtual connection between the port created with the free software “Virtual Serial Port Emulator”. Two 16F886 microcontrollers have been defined and used virtually so that one acts as a master sending data through I^2^C and the other as a slave to receive and display the received data. The combination of software used reduces the need for physical components as they can be programmed and debugged through simulation with the ISIS software.

## 5. Pulse Qualification

The pulse produced by NOPELED has been qualified. First, the electrical pulse generated by NOPELED has been measured. In addition, the pulse intensity has been measured, and, finally, the optical pulse for three models of LEDs with different wavelengths has been obtained.

### 5.1. Electrical Pulse Shape

The rise time and FWHM of the electrical pulse determine the shape of the optical pulse. The test setup used to measure both parameters is shown in Figure 6. The electrical signal in the terminals of the LED is derived from an 8-GHz oscilloscope, where it is captured at 10 GSPS. The electrical pulse is acquired for several voltages; the data are stored, treated, and plotted using Matlab software. In Figure 7, the electrical pulse measured in the pads of the LED is shown. The LED used in this test was 405 nm, the voltage used was 30 V, and the frequency was 1 kHz. As can be seen, after the electrical pulse, a negative oscillation follows. This is caused by the inductor located in parallel with the LED. The rise time is about 2.2 ns. The measurement has been repeated for different voltages (10, 15, 20, 25, and 30 V) and is presented in Figure 8, where the same measurement has been performed for two more models of LEDs flashing at 505 and 532 nm and without an LED. The same measurements, organized by LEDs, are presented in Figure 9. The rise time of the electrical pulses is presented in Figure 10. The rise times vary between 2.1 and 2.5 ns. The maximum value of the electrical pulse increases with the voltage applied. The variations from LED model to LED model are probably due to different impedance.

### 5.2. Pulse Intensity

The pulse intensity has been qualified in two different LEDs of the model HLMP-CM1A-560DD [35] at different frequencies and voltages. A Newport energy meter 1835-C holding the head (818-UV) has been used to measure the pulse intensity. In Figure 11, the data obtained for each LED at different frequencies (1, 2, and 4 kHz) and voltages (15, 20, 25, and 30 V) are plotted. As can be observed, the pulse intensity increases with the voltage while it stays almost constant, decreasing slightly with the frequency changes. The error of the setup is less than 1%.

### 5.3. Optical Pulse Shape

In order to assess the optical pulse generated by NOPELED, the SPE technique [7,36,37] has been employed. By using this technique, it is possible to reconstruct the pulse shape and obtain the rise time and FWHM. The time differences between the electrical signals generated by two photo-multipliers are accumulated in a histogram, with the histogram being the reconstruction of the pulse shape. Both photo-multipliers are illuminated by NOPELED. The first photo-multiplier works after attenuating the light emitted at the photo-electron level, and the second one, the trigger photo-multiplier, works at normal levels of light. The signal of the photo-multiplier working at the photo-electron level advances and delays depending on the part of the optical pulse from where the photon comes. Moreover, as the probability depends on the pulse shape, the histogram that accumulates the delays between the trigger and the photo-electron photo-multipliers reproduces the shape of the light pulse. The pulser is located in a dark box. The LED to be used is soldered to the pulser circuit. Then, the voltage and frequency parameters are adjusted from the console. Finally, the level of light in the photo-electron photo-multiplier is adjusted with filters in order to have about 1% of events. Once all the values are well adjusted, the measurement is started, taking data and generating the histogram. In Figure 12, the test setup can be observed. The readout lasts several hours until there are enough data in the histogram. The histogram represents the optical shape of the pulse. The skew between the electrical trigger signal (positive edge) and the trigger photo-multiplier (negative edge) has been measured (see Figure 13), and it has a standard deviation of 184 ps. To obtain a good fit of the histogram data, from where to measure the rise time and FWHM, Matlab software has been used. In this case, an adjustment to the Weibull function has been carried out.

Equation (Equation 1) describes the Weibull function used to fit the data:(1)$$f\left(x\right)=\alpha \cdot \left(\frac{\beta \cdot x^{\left(\beta -1\right)}}{\theta^{\beta }}\cdot e^{-{\left(\frac{x}{\theta }\right)}^{\beta }}\right)$$

Being: α = multiplicative factor; β = form factor; θ = scale factor.

One example of the pulse shape of NOPELED obtained with the photo-electron method is presented in Figure 14. As can be seen, a rise time of less than 3 ns is obtained. Further, the FWHM of the optical pulse is lower than 7 ns. The trigger frequency for this measurement is 1 kHz and the voltage 15 V, while the LED model used is a HLMP-CM1A-560DD, which emits at 532 nm. The factors of interest are the rise time and the FWHM time. To observe the variation in these values at different values of voltage and frequency of operation, several LEDs have been analyzed. The optical pulse has been obtained at different voltages (15, 20, 25, and 30 V) with a fixed frequency of 1 kHz for two LEDs of the models HLMP-CM1A-560DD and HLMP-CE34-Y1CDD [38], the latter emitting at 505 nm. The optical pulse has also been obtained for a fixed voltage of 30 volts and different pulsing frequencies (1, 2, 5, and 10 kHz) for the same LEDs. In addition, for one LED of the model VAOL-5EUV0T4 [39], the emission wavelength of which is 405 nm, the optical pulse for an extended range of frequencies (15, 20, 25, and 30 kHz) when operating the pulser at 20 V has been obtained. The results obtained are presented in Figure 15. As can be observed, the rise time increases with the voltage applied, while the variations in frequency do not seem to have an effect on the optical pulse. Further, it is appreciated that with the wavelength emission of the LED, there is a decrease in the rise time, with the LED emitting at 405 nm, the one with a lower rise time, and the LED emitting at 532 nm, the one with the higher rise time. For the 405 nm LED, rise times lower than 2 ns have been measured (1.98 ns). The results obtained are sensibly higher than state-of-the-art optical pulse generators, in which optical pulses have a duration of a few picoseconds, even of short duration. However, the complexity of NOPELED is sensibly lower, which makes it appropriate for those applications where simplicity is required and where the pulse width and intensity provided are enough.

## 6. Conclusions

The design and architecture of NOPELED have been presented, including the electronics and the embedded software. The system consists of two low-cost boards that allow for generating very short optical pulses. Rise times of less than 3 ns and FWHM lower than 8 ns have been obtained with NOPELED while using LEDs emitting at wavelengths ranging from 405 to 532 nm. In the particular case of the 405 nm LED, rise times lower than 2 ns have been obtained. NOPELED can modify the light emission intensity as well as the frequency. Finally, the qualification results have been presented, measuring and validating NOPELED to be used by those applications needing inexpensive and simple short optical pulse generators.

## Figures and Tables

**Figure 1 sensors-22-07683-f001:**
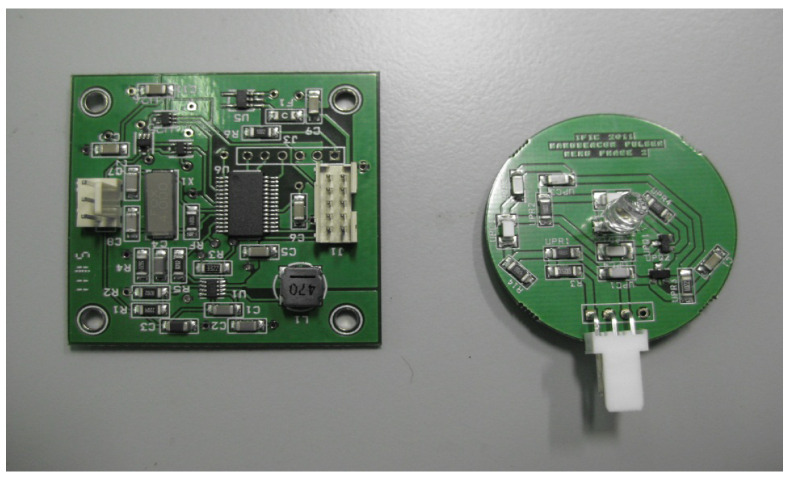
NOPELED Electronics boards: the controller and the pulser.

**Figure 2 sensors-22-07683-f002:**
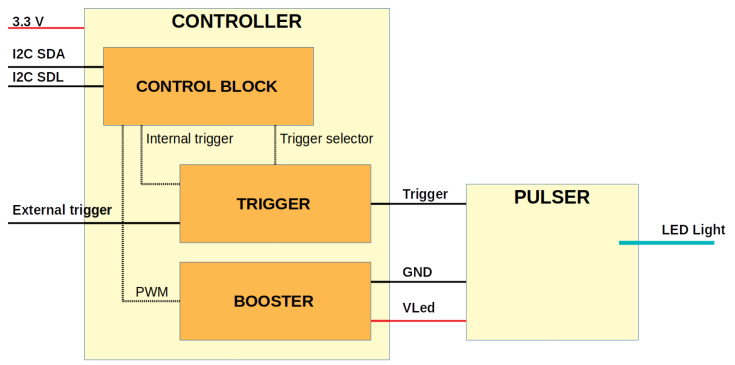
The architecture of NOPELED. The pulser board contains the switching electronics to make the LED flash. The control board includes the booster, where the variable voltage to supply the pulser is generated; the trigger block, where the internal or external trigger is selected; and the control block, where the internal trigger is generated, the trigger selection is controlled, as well where the PWM signal to control the booster voltage is created. The control block also implements communications via I^2^C.

**Figure 3 sensors-22-07683-f003:**
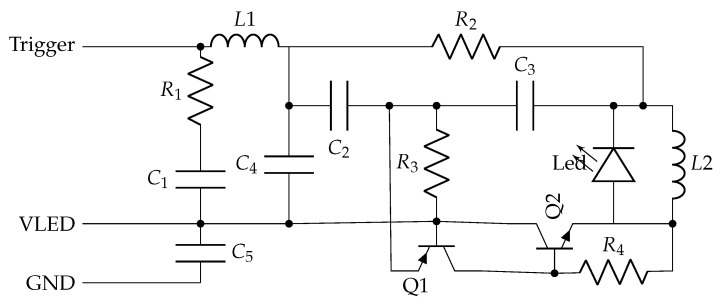
Pulser circuit. The values of the components are: L1 = L2 = 100 µH; C1 = 100 nF; C2 = 47 pF; C3 = 100 pF; C4 = 10 pF; C5 = 10 nF; R1 = 100 Ω; R2 = 100 kΩ; R3 = 2.2 kΩ; and R4 = 10 kΩ.

**Figure 4 sensors-22-07683-f004:**
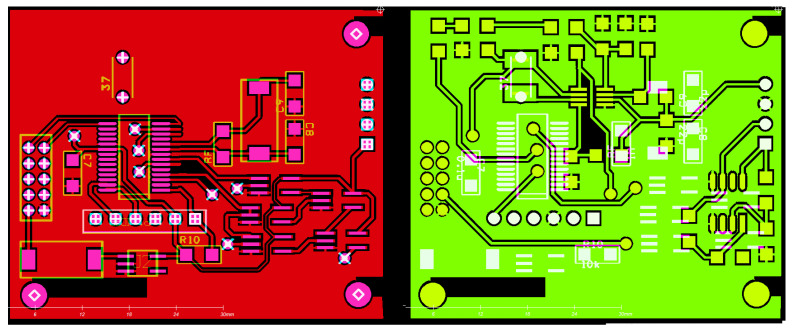
PCB layout of the NOPELED controller. Top and bottom layers of the control board.

**Figure 5 sensors-22-07683-f005:**
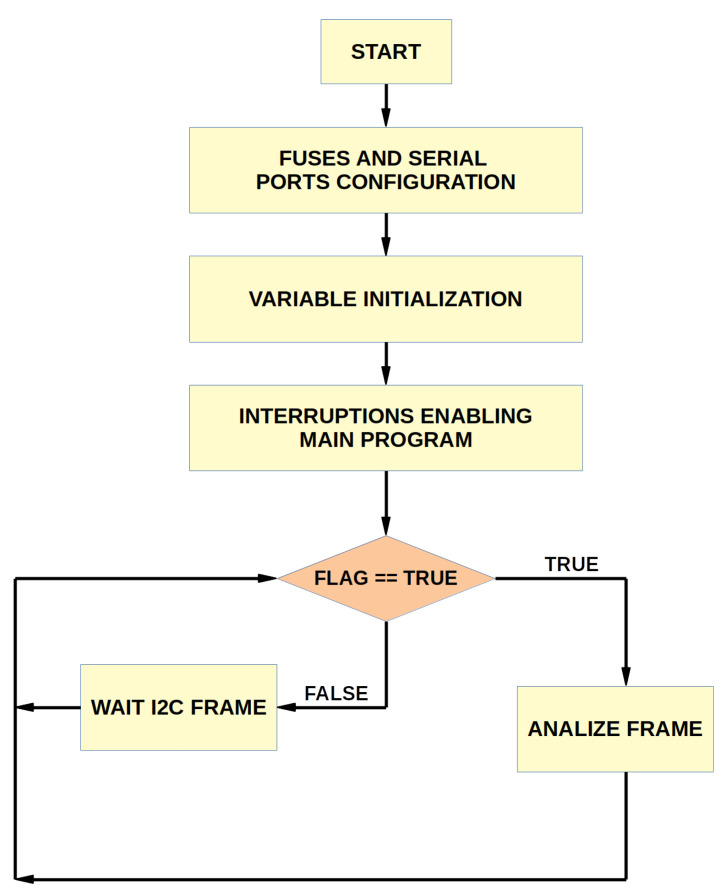
Flow diagram of NOPELED embedded software. At start-up, the fuse and serial configuration are performed. The next step is the initialization of the variables, which is followed by interrupt enabling. After these initial steps, the embedded software waits for the arrival of I^2^C messages. The received command is analyzed, and the corresponding tasks are performed, coming back to the waiting stage.

**Figure 6 sensors-22-07683-f006:**
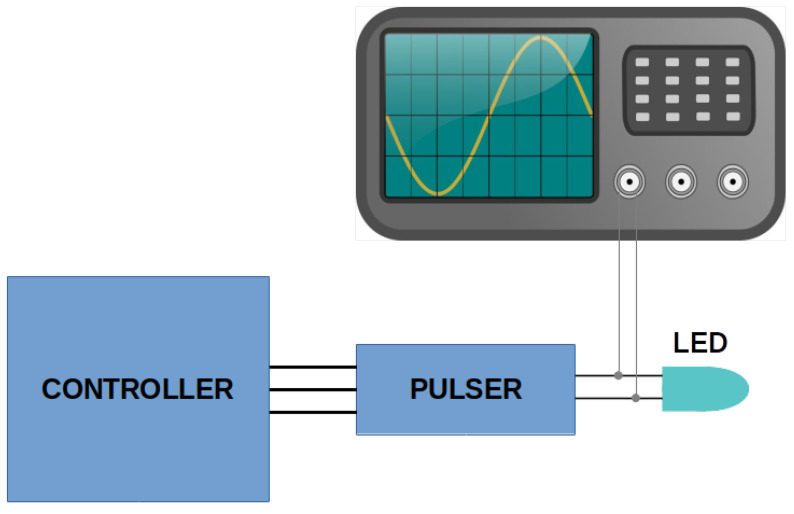
Tests setup for the measurement of the electrical pulse. The electrical pulse is measured by connecting the pads of the LED to one 8-GHz oscilloscope channel.

**Figure 7 sensors-22-07683-f007:**
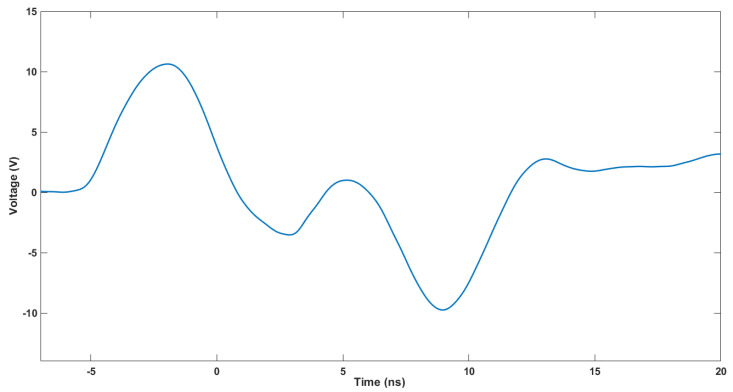
Electrical pulse measured with a 405 nm LED, supplied at 30 V with a frequency of 1 kHz. The negative after pulse, created by the inductor in parallel with the LED, helps to decrease the duration of the optical pulse.

**Figure 8 sensors-22-07683-f008:**
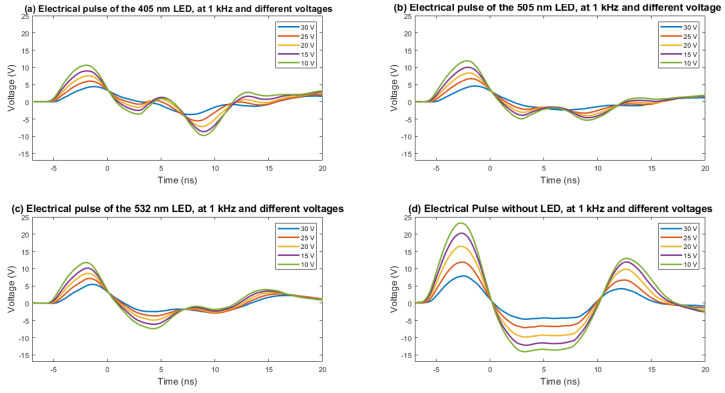
The electrical pulse is obtained with LEDs of 405 nm (**a**), 505 nm (**b**), and 532 nm (**c**). The electrical pulse without LED (**d**) is also presented. The pulses are presented for each LED, varying the voltage from 10 to 30 V in steps of 5 V and with a fixed frequency of 1 kHz.

**Figure 9 sensors-22-07683-f009:**
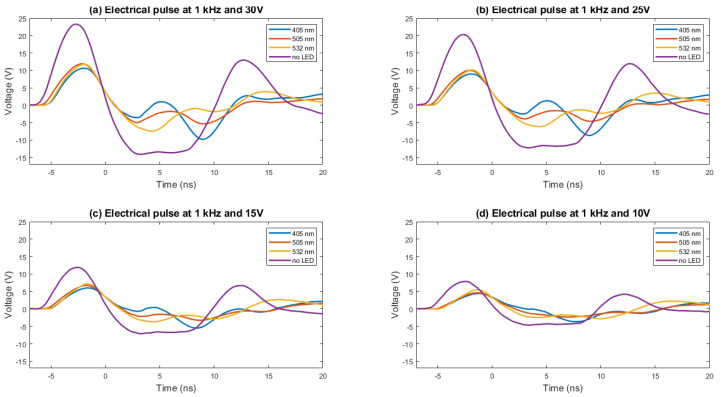
The electrical pulse is obtained with LEDs of 405, 505, and 532 nm. The electrical pulse without LED is also presented. The pulses in this case are presented by voltage (10 (**a**), 15 (**b**), 25 (**c**), and 30 V (**d**)). The impedance of the LED makes the system slow down. The differences in the electrical pulse between LEDs are also due to small differences in the LED impedance. The negative oscillation is caused by the inductor placed in parallel with the LED to reduce the pulse duration.

**Figure 10 sensors-22-07683-f010:**
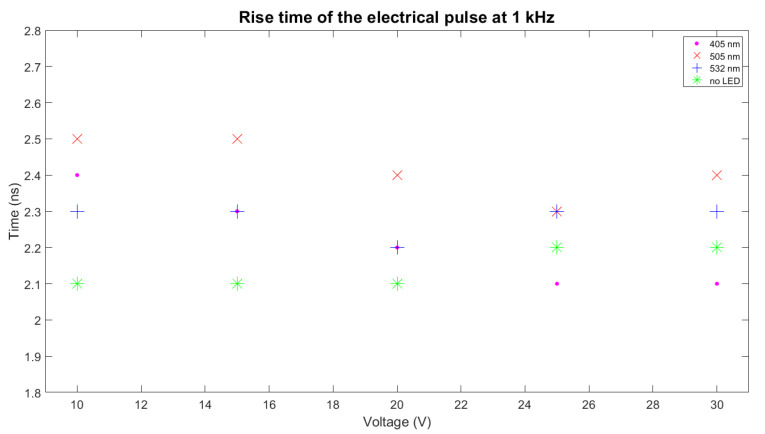
Rise time of the electrical pulse for 405, 505, and 532 nm LEDs. It is also presented the rise time without LED.

**Figure 11 sensors-22-07683-f011:**
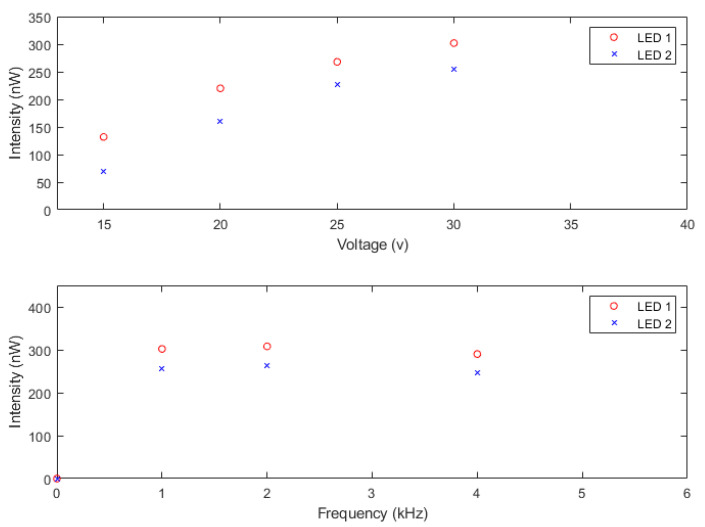
NOPELED pulse intensity measured in two LEDs at different values of voltage with the frequency fixed at 1 kHz (**top**). The pulse intensity has also been measured at different frequencies with the voltage fixed at 30 V (**bottom**). The intensity increases with the voltage and stays stable with frequency.

**Figure 12 sensors-22-07683-f012:**
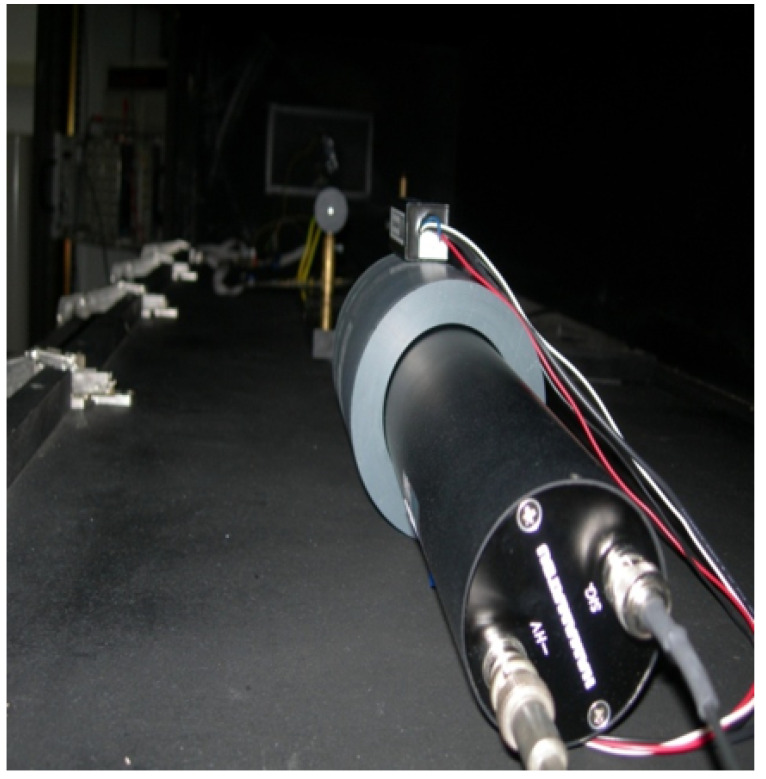
Photo-electron setup inside the dark box. In the foreground, the photo-electron PMT with the trigger PMT mounted on top it is seen. In the background, the LED pulser can be observed.

**Figure 13 sensors-22-07683-f013:**
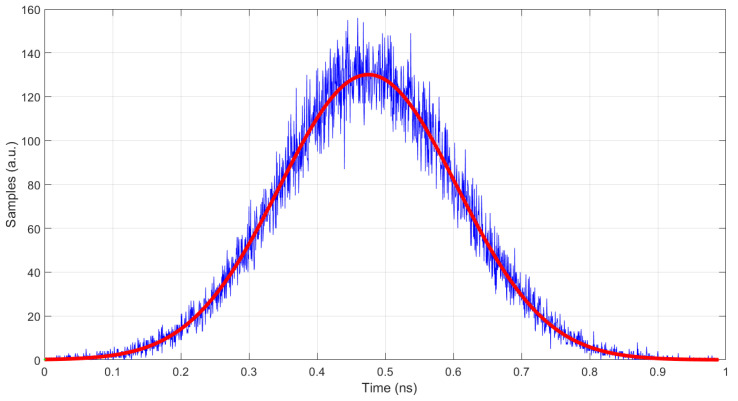
Skew between the electrical trigger and the trigger photo-multiplier. The data have been fit to a Gaussian function. The standard deviation is 184 ps.

**Figure 14 sensors-22-07683-f014:**
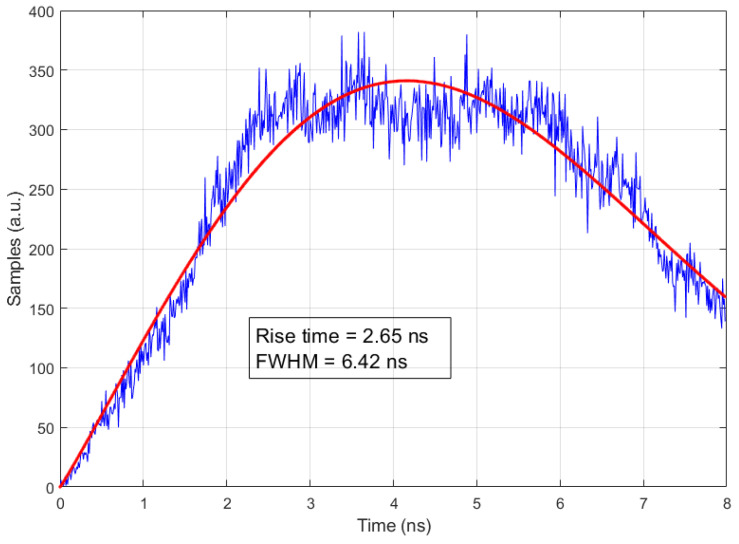
Pulse shape of NOPELED obtained using the technique of the photo-electron. NOPELED was operated at 1 kHz, and a voltage of 15 V. The histogram obtained has been fit to a Weibull function, obtaining a rise time of 2.65 ns and an FWHM of 6.42 ns.

**Figure 15 sensors-22-07683-f015:**
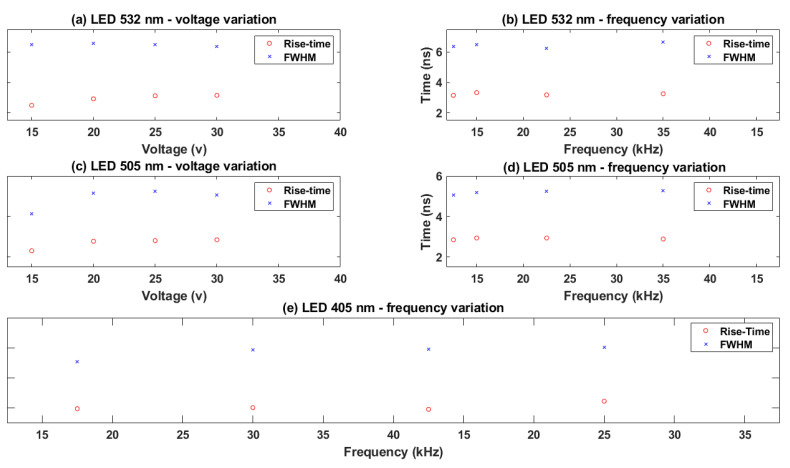
Optical pulse results for three different LED models at 532 (**a**,**b**), 505 (**c**,**d**), and 405 nm (**e**). The variation in the rise time and FWHM with respect to the voltage (**a**,**c**) and frequency (**b**,**d**,**e**) is presented for LEDs of 532 and 505 nm, while the variation with respect to frequency is presented for 405 nm.

## Data Availability

Data can be provided under request.

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
