# Peer review of "A Narrow Optical Pulse Emitter Based on LED: NOPELED"

_sensors, 2022, doi:10.3390/s22197683_

Round 1

Reviewer 1 Report

This manuscript presents NOPELED, a light source based on LEDs emitting short optical pulses with typical rise times of less than three ns and Full Width at Half Maximum lower than seven ns. The device is inexpensive and has the advantage of modifying the pulse frequency and intensity. Its use can range from photo-multiplier to scintillator counter calibration. I think the work is valuable, and that can be accepted for publication after a minor revision. The comments are as follows:

1. The authors discuss about optical pulse. I suggest the authors see the paper "Ultrafast Science to Capture Ultrafast Motions", Ultrafast Science, vol. 2021, Article ID 9765859, 2 pages, 2021. (https://doi.org/10.34133/2021/9765859). Please add more discussion on ultrafast pulse in introduction part.

2. I suggest that the authors compare the performance of NOPELED with other reported results.

3. Recently some SAs based on 2D materials can also generate optical pulses.  (such as: Photonics Research 8 (1), 78-90); Laser Photonics Reviews, 2022, 16(2), 2100255); ACS Appl. Mater. Interfaces 12 (28), 3175731763.) Can the authors give some comments.

4. The figure 9 is interesting, can the author give some physical description.

5. Whether the operation wavelength can be widely tunable from the current to other wavelength?

6. This manuscript is readable, but there are some grammar errors, which should be polished.

Author Response

  1. The authors discuss about optical pulse. I suggest the authors see the paper "Ultrafast Science to Capture Ultrafast Motions", Ultrafast Science, vol. 2021, Article ID 9765859, 2 pages, 2021. (https://doi.org/10.34133/2021/9765859). Please add more discussion on ultrafast pulse in the introduction part.

Done

  1. I suggest that the authors compare the performance of NOPELED with other reported results.

 Done

  1. Recently some SAs based on 2D materials can also generate optical pulses.  (such as: Photonics Research 8 (1), 78-90); Laser Photonics Reviews, 2022, 16(2), 2100255); ACS Appl. Mater. Interfaces 12 (28), 31757–31763.) Can the authors give some comments?

 Done

  1. The figure 9 is interesting, can the author give some physical description.

Added in the Figure caption

  1. Whether the operation wavelength can be widely tunable from the current to other wavelengths?

The operation wavelength cannot be tuned, it is fixed with the selection of the LED. This is clarified in the manuscript:  “While the wavelength is fixed for a given LED model, the intensity and the frequency of the optical pulse can be controlled”

  1. This manuscript is readable, but there are some grammar errors, which should be polished.

The full manuscript has been reviewed thoroughly

Reviewer 2 Report

This paper is not suitable for Sensors; there is not any sensing application to demonstrate the capability of the LED. This paper is more appropriate for an instrumentation journal. Additionally, the introduction part is very short and poor; there is no any mention to the state of the art publications related to the topic.

Author Response

The authors would like to thank the reviewer for the useful comments and suggestions.

This paper is not suitable for Sensors; there is not any sensing application to demonstrate the capability of the LED. This paper is more appropriate for an instrumentation journal. Additionally, the introduction part is very short and poor; there is no any mention to the state of the art publications related to the topic.

We consider that manuscript enters in the scope of “Optoelectronic and photonic sensors” as part of the calibration system of the overall sensor system, as the sensor qualification, and the calibration methods and instrumentation, improve the performances of the complete sensor system when they are not part of the sensorization system as well.  References have been provided for the use of short pulses of LED light, such as the one provided by NOPELED, in scintillation counter calibration, photomultiplier calibration, protein fluorescence, etc. Also an example of PMT calibration has been added to the text: the case of the time calibration of a neutrino telescope.

The introduction has been improved to consider the state-of-the-art on the topic.

Reviewer 3 Report

Dear Authors,

The manuscript should be improved and then could be resubmitted to Sensors.

I recommend considering of following comments:

1. The manuscript begins very well and ends up with mistakes (English, typing mistakes, quality of figures).

2. Page 3, Figure 3, Q1 should be corrected.

3. Page 6, Fig. 5 legend, grammar should be checked.

4. Page 7, the axes labels should be reversed.

5. Page 8, there is no need for figure 9. Figure 8 should have proper indices for each figure and proper legend to refer to it (use Fig. a, b, c, for better interpretation of presented data).

6. Page 9, Figure 10, the axes labels are not visible.

7. Page 9, Line 202 correct the sentence.

8. Page 10, Table 1 is not necessary.

9. Page 11, extensive English language corrections are needed.

10. Page 12, Correct the Figure 13 legend.

11. Page 12, the value mentioned in Figure 14 does not correspond with the value mentioned in referring text or in figure legend.

12. Figures 13 could be moved after the referring text.

13. Page 13, Figure 15, a higher font size of the text used in the figure should be employed (in the insets and axes labels).

14. Page 13, Figure 15, the Figure legend could be improved by adding appropriate data.

15. Page 13, line 250, the authors claim that "the rise time decreases with the voltage applied", however, from figure 15 this can not be clearly understood. Moreover, by looking at the data, by increasing the applied voltage a slight increase in the rise time could be observed for LEDs emitting at 532 nm and 505 nm.

16. Page 13, Figure 15, the voltage and/ or frequency should be specified in the legend. The figure 15 could be easily interpreted by using Fig 15 a, b, c, d, etc. The legend of Figure 15 should be completed.

17. Page 13, Line 239, the presented information should be checked.

Author Response

The authors would like to thank the reviewer for the thoughtful review of the manuscript and the useful corrections, comments and suggestions.

The manuscript begins very well and ends up with mistakes (English, typing mistakes, quality of figures).

  1. Page 3, Figure 3, Q1 should be corrected.

Done

  1. Page 6, Fig. 5 legend, grammar should be checked.

Done

  1. Page 7, the axes labels should be reversed.

Done

  1. Page 8, there is no need for figure 9. Figure 8 should have proper indices for each figure and proper legend to refer to it (use Fig. a, b, c, for better interpretation of presented data).

Added the references to the figures. Figure 9 has been kept as another reviewer finds it interesting.

  1. Page 9, Figure 10, the axes labels are not visible.

Corrected

  1. Page 9, Line 202 correct the sentence.

Done

  1. Page 10, Table 1 is not necessary.

           Removed

  1. Page 11, extensive English language corrections are needed.

Done

  1. Page 12, Correct the Figure 13 legend.

Done

  1. Page 12, the value mentioned in Figure 14 does not correspond with the value mentioned in referring text or in figure legend.

 Corrected

  1. Figures 13 could be moved after the referring text.

 Done

  1. Page 13, Figure 15, a higher font size of the text used in the figure should be employed (in the insets and axes labels).

 Done

  1. Page 13, Figure 15, the Figure legend could be improved by adding appropriate data.

 Done

  1. Page 13, line 250, the authors claim that "the rise time decreases with the voltage applied", however, from figure 15 this can not be clearly understood. Moreover, by looking at the data, by increasing the applied voltage a slight increase in the rise time could be observed for LEDs emitting at 532 nm and 505 nm.

Done: “The rise time increases with the voltage applied”. In fact, this is what we wanted to say, that the lower the voltage, the lower the rise time.

  1. Page 13, Figure 15, the voltage and/ or frequency should be specified in the legend. The figure 15 could be easily interpreted by using Fig 15 a, b, c, d, etc. The legend of Figure 15 should be completed.

           Done

  1. Page 13, Line 239, the presented information should be checked.

Done. Line modified to: The trigger frequency for this measurement is  1 kHz and the voltage 15 V, while the led model used is a HLMP-CM1A-560DD, which emits at 532 nm.

Round 2

Reviewer 2 Report

I agree with authors responses

Author Response

Dear reviewer,

   Thanks for your comments and suggestions.

Best regards

Reviewer 3 Report

Dear Authors,

Figure 3 and 9 legends should be verified and corrected.

Best regards

Author Response

Dear reviewer,

   Both legends have been corrected.

Kind regards,

Diego